# Probe exciplex structure of highly efficient thermally activated delayed fluorescence organic light emitting diodes

Tzu-Chieh Lin[1], Monima Sarma[1], Yi-Ting Chen[1], Shih-Hung Liu[1], Ke-Ting Lin[2], Pin-Yi Chiang[2], Wei-Tsung Chuang[3], Yi-Chen Liu[4], Hsiu-Fu Hsu[4], Wen-Yi Hung[2], Wei-Chieh Tang[1], Ken-Tsung Wong[1,5] & Pi-Tai Chou[1]

The lack of structural information impeded the access of efficient luminescence for the exciplex type thermally activated delayed fluorescence (TADF). We report here the pump-probe Step-Scan Fourier transform infrared spectra of exciplex composed of a carbazole-based electron donor (CN-Cz2) and 1,3,5-triazine-based electron acceptor (PO-T2T) codeposited as the solid film that gives intermolecular charge transfer (CT), TADF, and record-high exciplex type cyan organic light emitting diodes (external quantum efficiency: 16%). The transient infrared spectral assignment to the CT state is unambiguous due to its distinction from the local excited state of either the donor or the acceptor chromophore. Importantly, a broad absorption band centered at ~2060 cm$^{-1}$ was observed and assigned to a polaron-pair absorption. Time-resolved kinetics lead us to conclude that CT excited states relax to a ground-state intermediate with a time constant of ~3 μs, followed by a structural relaxation to the original CN-Cz2:PO-T2T configuration within ~14 μs.

[1] Department of Chemistry, National Taiwan University, Taipei 10617, Taiwan. [2] Institute of Optoelectronic Sciences, National Taiwan Ocean University, Keelung 202, Taiwan. [3] National Synchrotron Radiation Research Center, Hsinchu City 30076, Taiwan. [4] Department of Chemistry, Tamkang University, New Taipei City 25137, Taiwan. [5] Institute of Atomic and Molecular Science, Academia Sinica, Taipei 10617, Taiwan. Correspondence and requests for materials should be addressed to W.-Y.H. (email: wenhung@mail.ntou.edu.tw) or to K.-T.W. (email: kenwong@ntu.edu.tw) or to P.-T.C. (email: chop@ntu.edu.tw)

Lighting materials with thermally activated delayed fluorescence (TADF) have been attracting considerable attention in the organic light emitting diodes (OLEDs)[1–4]. The core of TADF molecules lies in their small singlet–triplet energy splitting, $\Delta E_{\text{T}-\text{S}}$ ($T_1 - S_1$ in energy), which leads to efficient repopulation from the emission forbidden triplet states ($T_1$) to the allowed singlet states ($S_1$) via thermal activation[1,5–8]. To fulfill this criterion, the correlation of two open shell electrons in the triplet state has to be reduced, so that the electron exchange energy can be minimized. This requires large separation and hence weak coupling between two adjacent singly occupied molecular orbitals. In one approach, the molecular identity can comprise an electron donor (D) and an electron acceptor (A), for which the highest occupied molecular orbital (HOMO) of donor and the lowest unoccupied molecular orbital (LUMO) are spatially separated. Upon photoinduced charge transfer (CT), HOMO and LUMO are virtually not or limited overlapped, leading to the reduction of electron exchange energy and hence a small $\Delta E_{\text{T}-\text{S}}$[9]. OLEDs based on intramolecular TADF emitters with >30% external quantum efficiencies (EQE) have been reported[10,11]. However, the challenges for the stability of TADF-based OLED devices have just begun[12–14]. The sophisticated molecular designs complicate the syntheses, which might hamper advances in optimization.

In another approach, via simply mixing D and A, the intermolecular CT, and hence TADF generation, showed prospects for the future due to its versatility in providing various combinations of D and A materials extensively developed for their use in hole- and electron-transporting materials, respectively[15]. The recent usage of exciplex-type host for phosphorescent/fluorescence OLEDs becomes popular[16]. The associated energy transfer is a Förster type, so that the doped fluorescent emitter can be in a very low concentration, rendering advantage for OLEDs on lower turn-on voltage and smaller roll-off[17]. Recently, Adachi and coworkers reported organic long persistent luminescence based on intermolecular TADF molecules, which is also expected to trigger the blossoms of new intermolecular TADF systems[18]. However, this facile approach still encounters challenges in accessing highly efficient exciplex systems[19–21]. One of the fundamental issues lies in the lack of structural information on the D/A exciplex. Unfortunately, the structure of such a purported exciplex remains elusive today, and that lack of knowledge impedes advances in exciplex-relevant OLEDs.

We report herein the pump-probe Step-Scan Fourier transform IR spectra of exciplex composed of a new carbazole-based electron donor (D: CN-Cz2) and a well-known 1,3,5-triazine-based acceptor (A: PO-T2T)[22], which are codeposited as a solid film. The cyano group on the CN-Cz2 offers two advantages. First, the -CN strong vibrational absorption at 2200–2500 cm$^{-1}$ serves as a marker for transient Fourier-transform infrared spectroscopy (FTIR). Secondly, the electron-withdrawing feature of CN group can subtly modulate the energy levels of donor and hence the exciplex emission.

## Results
### Structure characterization and photophysical property.
The cyano-contained donor CN-Cz2 (Fig. 1a) was synthesized and characterized with an X-ray structure (Supplementary Figs. 1, 2, and Supplementary Table 1). Basic characterizations indicate that CN-Cz2 exhibits good thermal/morphological stabilities suitable for vacuum deposition and bipolar charge-transporting character (Supplementary Figs. 3 and 4) as the -CN group was incorporated. The bipolar feature of CN-Cz2 is supported by the density function theory (DFT) calculation, revealing that HOMO resides on the peripheral carbazole, while LUMO is located on the CN-substituted carbazole core (Fig. 1b).

As shown in Fig. 1c, the absorption spectrum of the CN-Cz2:PO-T2T (1:1 weight ratio) exciplex system was identical to the combination of pristine CN-Cz2 and PO-T2T films, inferring negligible intermolecular interaction for the mixed films in the ground state. The PL spectrum was significantly red-shifted relative to both of the constituent molecules, correlating well with the energy difference between HOMO of CN-Cz2 (−5.70 eV determined by photoemission spectrometer, Supplementary Fig. 5) and LUMO of PO-T2T (−2.83 eV). The PLQY ($\Phi_{\text{PL}}$) for CN-Cz2:PO-T2T film was measured to be 55% in air (Supplementary Fig. 6). Figure 1d shows the decay curves of CN-Cz2:PO-T2T from the prompt emission ($\tau_1 = 35.68$ ns) to the end of delayed fluorescence emission ($\tau_2 = 3.39$ μs) at 300 K.

The mixtures of CN-Cz2:PO-T2T were introduced as EML in a device structure shown in Supplementary Fig. 7. The corresponding device based on CN-Cz2:PO-T2T (1:1) blend revealed a low turn-on voltage of 2.3 V and a maximum $L_{\text{max}}$ of 77,120 cd m$^{-2}$ at 10.0 V (4152 mA cm$^{-2}$) with CIE coordinates of (0.20, 0.40) (see Fig. 1e, f). The maximum efficiency reached 16% (EQE), 37.8 cd A$^{-1}$ (CE), and 47.5 lm W$^{-1}$ (PE). At a high luminance of 1000 cd m$^{-2}$, the device retained a high efficiency of 14.7% (EQE) at 3.4V, which is record-high performance among the exciplex-type cyan OLEDs[21]. The OLED performances with different CN-Cz2:PO-T2T ratios are showed in Supplementary Fig. 8 for which the EQEs and EL spectra were rather comparable to those of CN-Cz2:PO-T2T (1:1) (Fig. 1e–g, Supplementary Fig. 8 and Supplementary Table 2). In all cases, upon optical or electrical excitation, the energy transfer thus appeared to be very efficient due to the close proximity between CN-Cz2 and PO-T2T such that fast electron transfer takes place, resulting in CT TADF and hence highly efficient OLEDs.

### Step-Scan Fourier transform technology probing exciplex structure.
We then employed an unprecedented approach in attempts to probe the exciplex structure that remains elusive for all TADF molecules. To achieve this aim, we applied time-resolved IR spectroscopy using Step-Scan Fourier transform technology. Details of the experimental set-up and methodology are elaborated in Supplementary Fig. 9. In brief, a sample containing a mixture of CN-Cz2 and PO-T2T in various molar ratios was vacuum deposited on the CaF$_2$ substrate to prepare a film. A fourth harmonic (266 nm) of an Nd:YAG laser was used as the pump pulse, which was synchronized with the Step-Scan FTIR spectrometer and overlapped with the IR probe beam at an angle of ~45°. The sample thickness and roughness were crucial; the thickness was typically maintained at ~1 μm to have absorbance of ~5–10 at 266 nm. The roughness was kept at ±10 nm to avoid spatial variation.

We first performed a steady state IR measurement for CN-Cz2, PO-T2T, and CN-Cz2:PO-T2T (1:1). The results shown in Supplementary Fig. 10 provide clear evidence that the IR spectrum of the CN-Cz2:PO-T2T (1:1) mixture was identical to the sum of individual CN-Cz2 and PO-T2T IR spectra, indicating that simply mixing CN-Cz2 and PO-T2T in a 1:1 molar ratio caused negligible intermolecular interactions in the ground state, consistent with the conclusion made in early UV–vis absorption measurements. We then acquired the transient FTIR spectra of CN-Cz2, PO-T2T, and CN-Cz2:PO-T2T (1:1) with a pump (266 nm)-probe delay time of up to 60 μs. As a result, the characteristic transient IR spectra were acquired for CN-Cz2, PO-T2T, and CN-Cz2:PO-T2T exciplex, and they are shown in Fig. 2a (1200–1700 cm$^{-1}$) and b (1700–3000 cm$^{-1}$) at a delay time of 1.6 μs. The vibrational peaks showing positive absorbance were

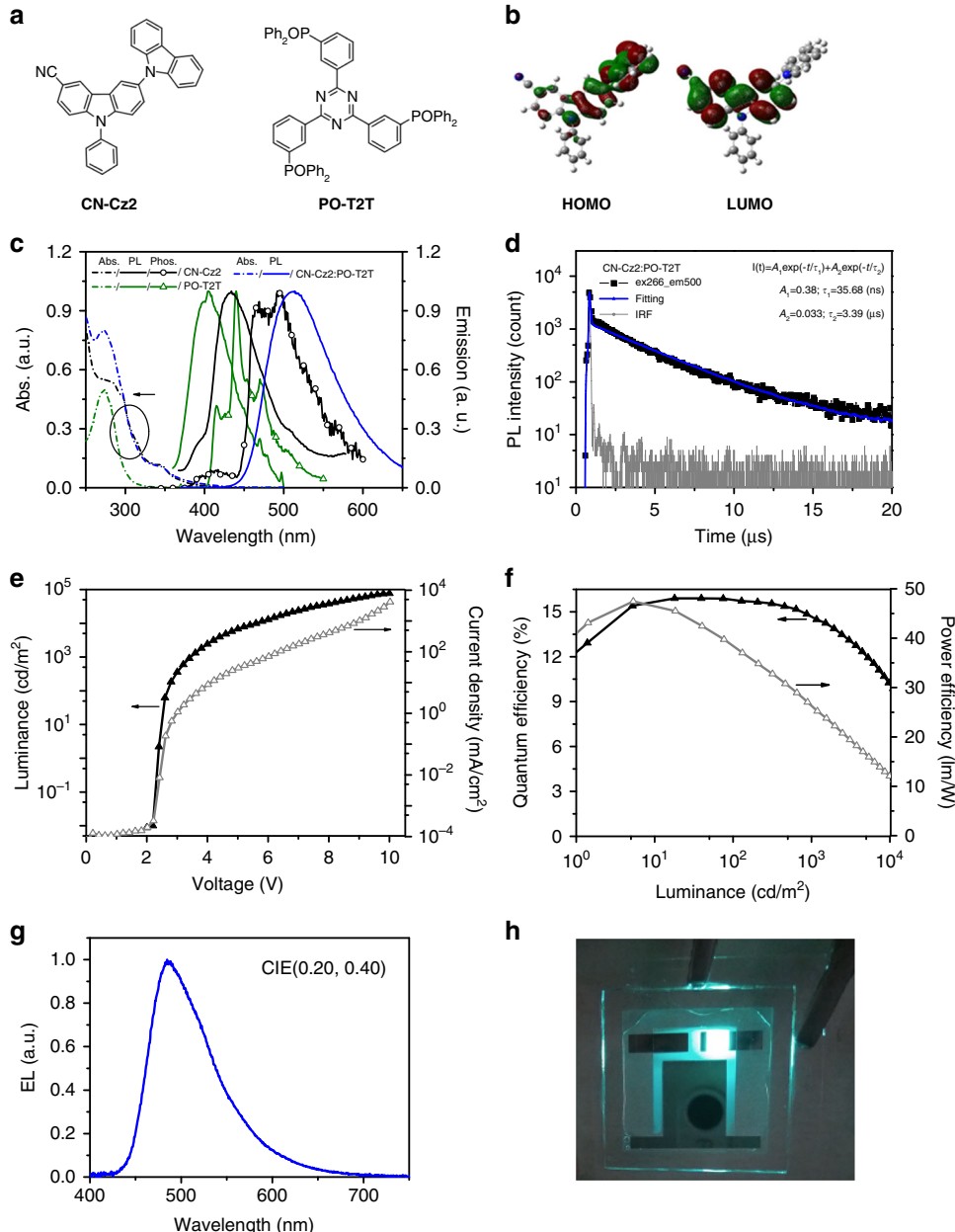

**Fig. 1** Photophysical properties of CN-Cz2, PO-T2T, and the exciplex CN-Cz2:PO-T2T (1:1). **a** Chemical structures of CN-Cz2 and PO-T2T. **b** Highest occupied molecular orbital (HOMO) and lowest unoccupied molecular orbital (LUMO) of CN-Cz2 according to DFT calculations at the B3LYP/6-31G(d) level. **c** Normalized photoluminescence (PL) and absorption spectra of CN-Cz2, PO-T2T as well as their blend film in the ratio 1:1. **d** Transient PL decay properties of CN-Cz2:PO-T2T (1:1) blend film at 300 K. The performance of OLEDs using CN-Cz2:PO-T2T (1:1) blend film as the emitter. **e** J–V–L characteristics. **f** EQE–PE–L characteristics. **g** EL spectrum of the CN-Cz2:PO-T2T (1:1) exciplex-based OLED. **h** A prototype demonstration of CN-Cz2:PO-T2T (1:1) exciplex OLED lighting device

ascribed to the IR spectra of CN-Cz2, PO-T2T, and CN-Cz2:PO-T2T (1:1) in the excited state. Those vibrational peaks having negative absorbance were accordingly associated with IR spectra of the ground-state recovery. Note that, due to the limitations on laser duration, jittering and system response, the early dynamics of a few tens of nanoseconds could not be resolved. Therefore, the acquired temporal IR spectral evolution here was mainly associated with the long-lived (>50 ns) transient species. For individual CN-Cz2 and PO-T2T, due to their lack of any TADF, the observed μs transient IR peaks should be ascribed to those of the local excited (LE) state in the triplet manifold. As for the transient IR spectrum of CN-Cz2:PO-T2T, careful analyses of Fig. 2a, b indicated that the transient IR peaks (with positive

absorbance) of CN-Cz2:PO-T2T could be ascribed to neither CN-Cz2 (or PO-T2T) nor the sum of CN-Cz2 and PO-T2T. Several peaks appearing at, e.g., 1319, 1547, and 1570 cm$^{-1}$ in CN-Cz2:PO-T2T (1:1) were new, and they were distinct from those of either CN-Cz2 or PO-T2T. Detailed transient IR data are summarized in Supplementary Table 3.

The 3D plots (absorbance–frequency–time) for CN-Cz2:PO-T2T (1:1) transient IR spectra are depicted in Fig. 2c. Upon monitoring at the newly appearing peak, such as 1570 cm$^{-1}$, as shown in Fig. 2d, the decay time constant was fitted to be 3.3 ± 0.2 μs, which, within experimental error, is consistent with the 3.39 μs time constant of delay fluorescence acquired by TCSPC (Fig. 1d). Similar relaxation dynamics were obtained for other IR

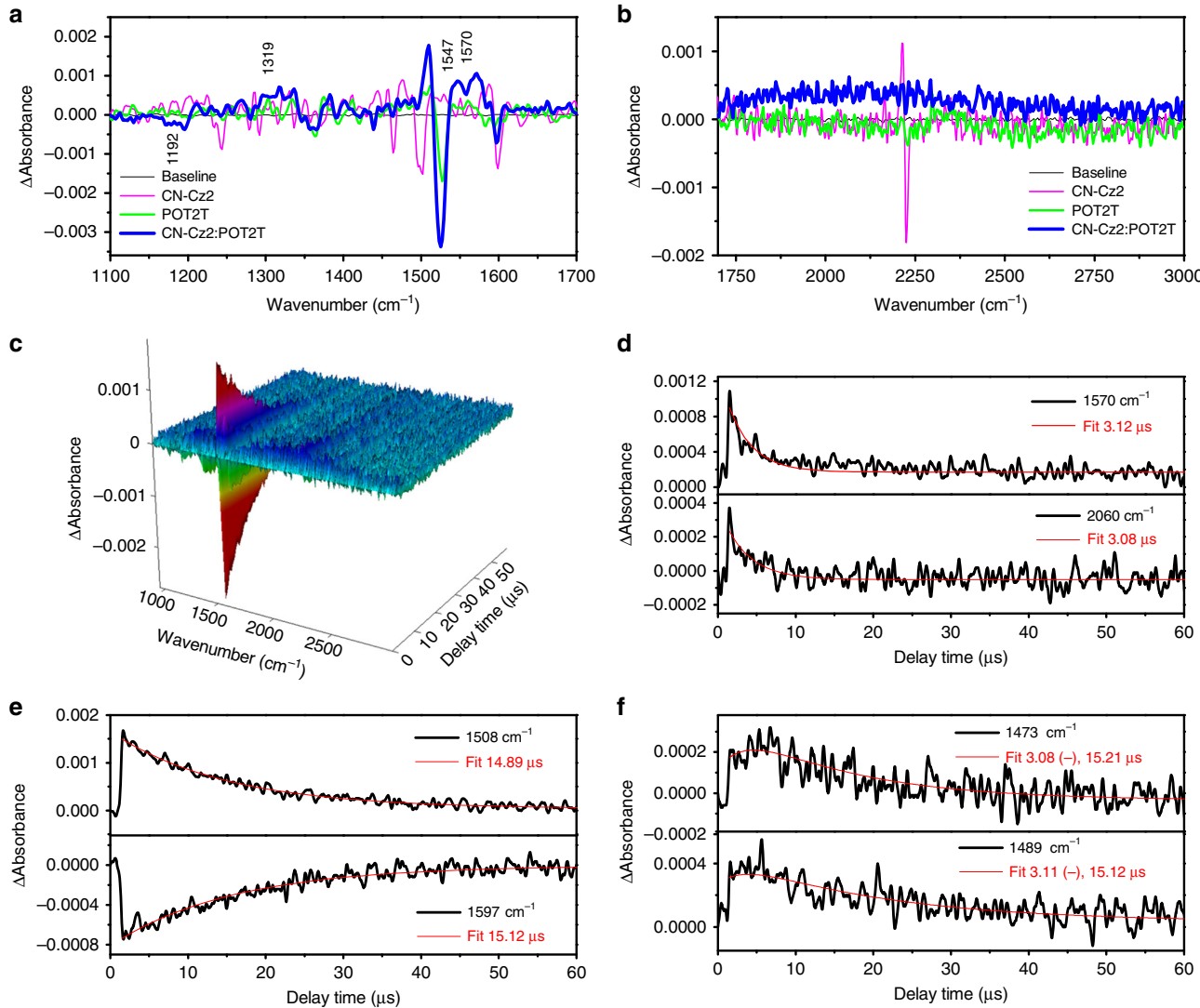

**Fig. 2** The transient IR character of thin film of CN-Cz2, PO-T2T, and CN-Cz2:PO-T2T (1:1). The transient IR spectra of these three compounds at a delay time of 1.6 μs in different spectral range of (**a**) 1200–1700 cm$^{-1}$ and (**b**) 1700–3000 cm$^{-1}$. **c** An absorbance-frequency-time 3D plot for the CN-Cz2:PO-T2T (1:1). **d** The relaxation dynamics of 1570 and 2060 cm$^{-1}$, (**e**) 1508 and 1597 cm$^{-1}$, and (**f**) 1473 and 1489 cm$^{-1}$ peaks for the CN-Cz2:PO-T2T (1:1) film. $\lambda_{ex}$: 266 nm

peaks, such as 1319 and 1547 cm$^{-1}$ (see Supplementary Fig. 11). Therefore, the resultant transient IR for CN-Cz2:PO-T2T (1:1) cannot be ascribed to the LE state of either donor (CN-Cz2) (Supplementary Fig. 12) or acceptor (PO-T2T) (Supplementary Fig. 13), but rather the CT state of the CN-Cz2:PO-T2T. Because of fast interconversion, the associated peaks were indistinguishable between S$_1$(CT) (delay component) and T$_1$(CT) states, which are also subject to mutual cancellation of intensity due to the rise-decay relationship. This makes full IR spectral assignment impractical.

Perhaps the most significant result for the transient IR of CN-Cz2:PO-T2T (1:1) lies in the observation of a broad band extending from 1700 to 3000 cm$^{-1}$ centered at ~2060 cm$^{-1}$ (Fig. 2b). This broad band, which is reproducible and does not appear in the pure CN-Cz2 or PO-T2T film, is authentic and undergoes similar relaxation dynamics (3.08 μs, see Fig. 2d) with that of the IR peaks of the exciplex. We tentatively assigned it to the polaron-pair absorption band. Polaron-pair absorptions have been reported to explain the IR ~2000–4000 cm$^{-1}$ broad absorption in conjugated donor/acceptor polymers that undergo electron–hole generation when voltage is applied[22–24]. In our

study here, the photoinduced electron transfer caused charge localization at CN-Cz2 (cation radical) and PO-T2T (anion radical) weakly bound by intermolecular static force[25], so the charge carrier was subject to significant polarization, generating polaron-pair (see Supplementary Fig. 14 for illustration). This characteristic polaron-pair absorption should be universal for the exciplex-type TADF emitters. Evidenced is given by the similar broad 1700–3000 cm$^{-1}$ transient IR spectra of CN-Cz2:PO-T2T (1:2) and CN-Cz2:PO-T2T (2:1) solid films (see Supplementary Fig. 15). More general, the polaron-pair absorption was observed for other types of exciplexes, for example the NPNPB (donor) and PO-T2T (see Supplementary Fig. 16c), in which a broad transient polaron-pair absorption appears and extends to >3000 cm$^{-1}$. NPNPB/PO-T2T has been reported to give inferior exciplex emission yield and OLED performance (EQE = 0.2%)[22]. For NPNPB/PO-T2T the results of higher polaron-pair absorption frequency than that of CN-Cz2:PO-T2T may indicate the more stable and hence higher efficiency of the polaron-pair formation that is subject to the subsequent polaron dissociation, resulting in the quench of emission. This trend may offer us a clue to explain the lower emission yield in the exciplex-type TADF

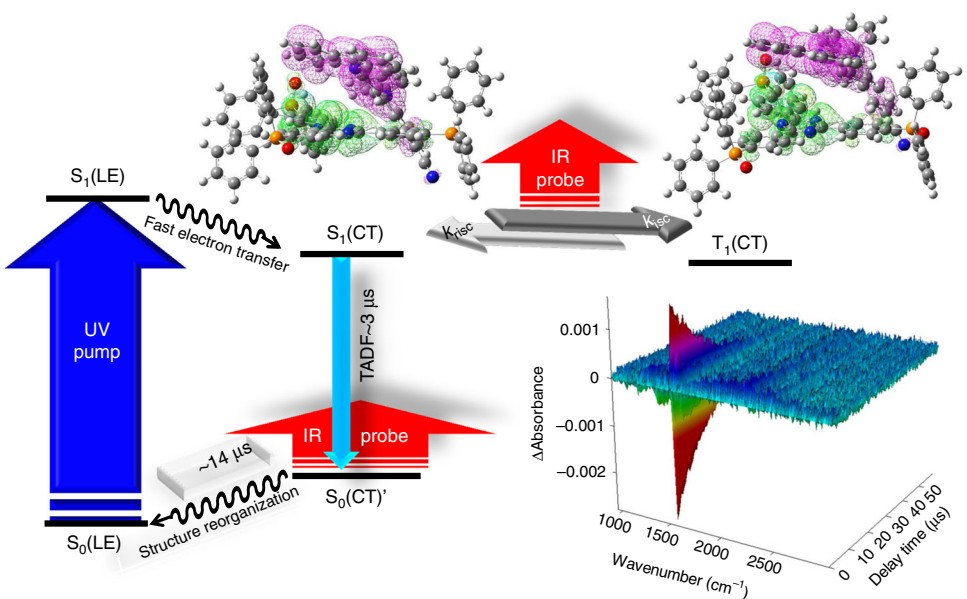

**Fig. 3** The proposed TADF cycle. There illustrate the pump (266 nm)-probe (IR) Step-Scan FTIR experiment combined with the theorical calculation structure: the optimized CN-Cz2:PO-T2T exciplex structure in its $S_1$ and $T_1$ state, associated with HOMO (pink) and LUMO (green). The isovalue for the contours is 0.02. Note that the distance shown in double arrow bar (red) is estimated by the distance between two center planes. See text for detailed explanation

molecules in common. Further support of this viewpoint is given by the comparative transient FTIR study of a number of highly emissive intramolecular TADF cases. As shown in Supplementary Figs. 17–21, our results indicate that there are none of the intramolecular TADF molecules for which the polaron-pair absorption could be observed in the range of 1700–3000 cm$^{-1}$. This is rational because the strong bound excitons lead to negligible polaron-pair formation in the intramolecular TADF cases.

Upon monitoring the rise kinetics of the negative peaks (see e.g., 1597 cm$^{-1}$ in Fig. 2e and Supplementary Fig. 22 for others), the recovery time constant is fitted to be 14.8 ± 0.3 μs, which is much longer than that of ~3.3 μs measured by both TADF and the transient IR peaks associated with $S_1$(CT) and/or $T_1$(CT) states (vide supra). We thus carefully scrutinize those positive IR bands that are in proximity of the negative IR peaks and found that they undergo similar ~14 μs relaxation time constant but with a decay behavior. For example, the kinetics of positive 1508 cm$^{-1}$ peak show a decay time of 14.9 μs (see Fig. 2e), which, within experimental error, is the same as the ground-state recovery time (15.1 μs) for the 1597 cm$^{-1}$ negative IR peak. In some positive IR absorbance regions such as 1473 and 1489 cm$^{-1}$, their relaxation dynamics reveal both rise and decay kinetics, which are fitted to be 3.3 ± 0.3 and 14.5 ± 0.3 μs, respectively (Fig. 2f, see Supplementary Fig. 22 for other peaks, Supplementary Table 3 for the complete list of analyzed peaks).

The observation of rise and decay components for the positive transient IR peaks leads us to propose the existence of an intermediate during an overall TADF cycle. Since the 14 μs decay component was not observed in both TADF and transient $S_1$(CT)/$T_1$(CT) IR spectra, the formation of intermediate is proposed to be in the ground state. Moreover, owing to a rise time of ~3.3 μs that is identical with the TADF decay time, the intermediate should be populated via the TADF process. As shown in Fig. 3 and Supplementary Fig. 14, upon forming exciplex in the $S_1$(CT) state, the charge induction may reorganize the CN-Cz2 (cation radical) and PO-T2T (anion radical) exciplex configuration for energy minimization. Accordingly, the recombination of polarons (see Supplementary Fig. 14) is expected to be

under a structure different from the original prepared $S_1$(CT), which upon vertical Franck–Condon transition, gives $S_0$(LE)' in the ground state. The $S_0$(LE)' state is then subject to further structure relaxation back to the initial CN-Cz2:PO-T2T configuration (see Fig. 3). It is reasonable to expect that the structural relaxation is imposed by steric interaction, inducing a non-negligible barrier, which rationalizes a time constant of 14 μs for the ground-state recovery. Evidence of structural relaxation may be provided by the prompt (delay = 0, gate width = 10 ns) and delayed (delay = 10 μs, gate width = 2 μs) components of PL and EL spectra for the CN-Cz2:PO-T2T film at 300K (see Supplementary Figs. 23 and 24), in which a red shift of peak wavelength of ~8–15 nm were resolved for the delayed components (cf. the prompt component). The spectral difference between prompt and delayed fluorescence has been commonly reported for the exciplex type TADF, while its origin is still pending[26–28]. Our transient IR results are in favor of the structural relaxation.

**Grazing incident X-ray diffraction study**. To gain in-depth insight into the exciplex formation, we have performed the grazing incident X-ray diffraction (GIXD) study to probe CN-Cz2 and PO-T2T codeposition film, so that the possible grain domain can be assessed. Deduced from GIXD for the CN-Cz2 and PO-T2T (1:1) codeposited film, Figure 4 indicates that the CN-Cz2 grain domain mainly consists of an average of 5 × 2 × 2 number of molecules where CN-Cz2 are packed along both in-plane (2 × 2 molecules) and out-of-plane (5 stacked molecules) orientations. For PO-T2T, no in-plane packing arrangement can be observed. Instead, the main alignment is along the out-of-plane orientation, which consists of an average of 15 PO-T2T molecules to make a rod-like configuration. With the deconvolution of GIXD signals (Supplementary Figs. 25 and 26), the signal shifts of the code-posited film GIXD can be deduced. It is clear that the interplanar spacings of the two components in the codeposited film are both larger than those in films of individual components, indicating interaction of the two components in the codeposited film. We further figured out that interaction between CN-Cz2 and PO-T2T is mainly based on the grain domain between CN-Cz2 and PO-

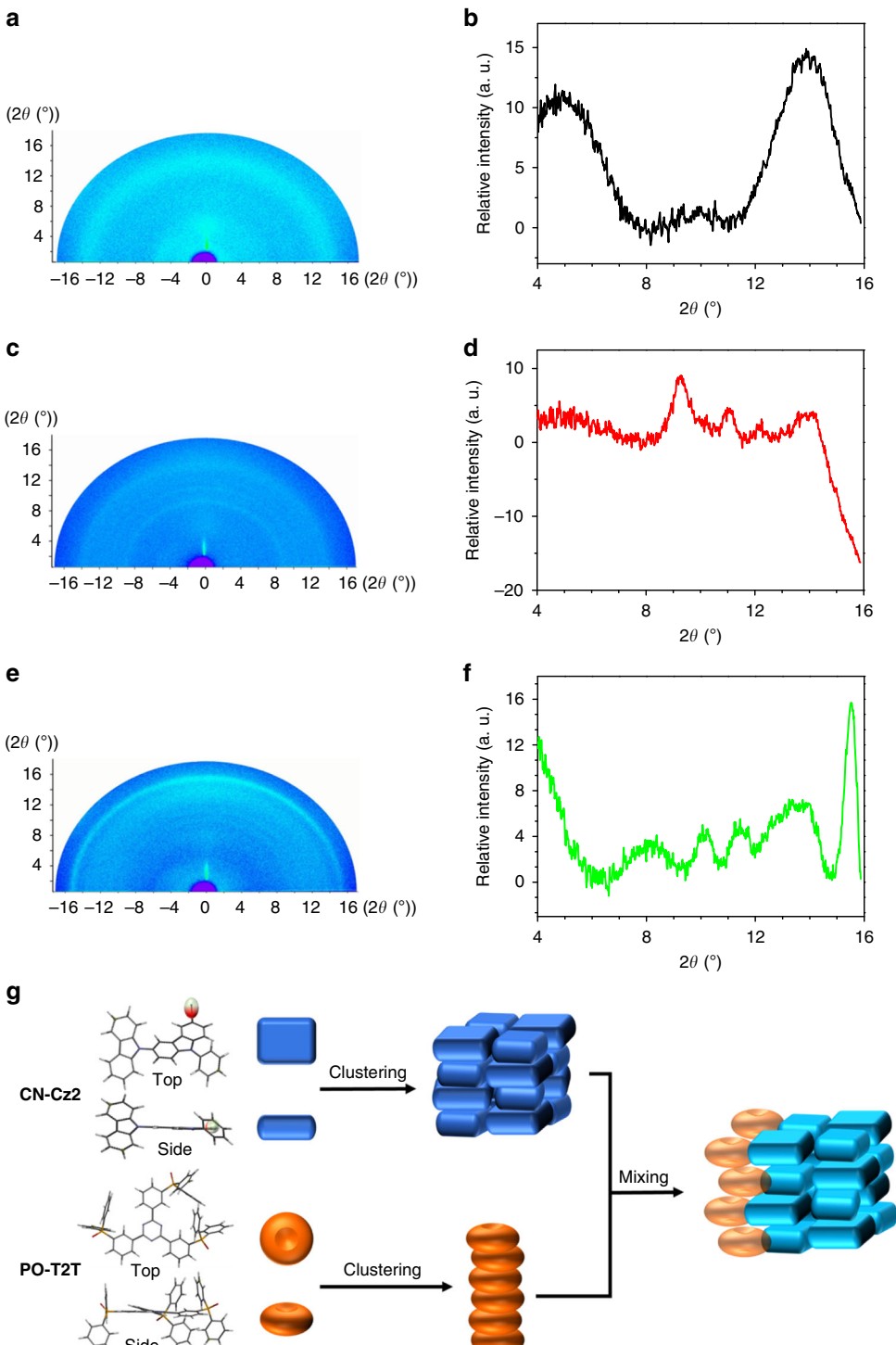

**Fig. 4** The GIXD character. 2D GIXD patterns for (**a**) CN-Cz2, (**c**) CN-Cz2:PO-T2T (1:1), (**e**) PO-T2T and respective 1D GIXD patterns (**b**, **d**, **f**). **g** The molecular packing for CN-Cz2 grain domain, PO-T2T grain domain, and the in-plane interaction between CN-Cz2 and PO-T2T grain domains. Note that the 1D GIXD profile of PO-T2T shows a strong diffraction peak at $2\theta = 15.5°$, corresponding to a well correlated π–π stacking with a d-spacing of 3.8 Å as illustrated in **g**. In CN-Cz2:PO-T2T (1:1) codeposited film, the diffraction corresponding to the π–π stacking of PO-T2T shifts to larger d-spacing as indicated by the deconvolution analyses. The interplanar spacings of the two components in the codeposited film are both larger than those in films of individual components, indicating interaction of the two components in the co-deposited film as depicted in **g**

T2T in an in-plane orientation (Fig. 4). The results lead us to propose that the contact between CN-Cz2 and PO-T2T, at the molecular level, is in an in-plane intercalation-like configuration, in which the polar–polar interactions between donor and acceptor sites plays a key role, rather than the π–π stacking interactions. Upon exciting the donor (or acceptor) moiety, the generated exciton, before its recombination, diffuses along the donor (or acceptor) grain domain. During the journey the excitons may encounter the D/A interface, from which the fast CT takes place if D/A is in an optimum configuration (see Supplementary Fig. 14). Otherwise, even excitons reach the D/A inactive boundary, CT won't occur. The right D/A configuration for CT,

based on numerous reports on intramolecular CT TADF molecules, may lie in rather small but nonnegligible overlap between HOMO of D (CN-Cz2) and LUMO of A (PO-T2T) to induce fast CT, so that the resulted CT state possesses closer (thermally accessible) energy between $S_1$ and $T_1$ states.

**Theoretical calculation**. We then performed the ab initio computational approach. Our methodology for calculation to obtain the putative exciplex configuration is elaborated in the supporting information. According to the above in-plane intercalation-like configuration, we considered two displaced orientations for CN-Cz2 and PO-T2T exciplexes, for which the -CN group in CN-Cz2 is toward or away from PO-T2T, specified as type I and II interaction, respectively (Supplementary Fig. 27). As a result, the optimized exciplex structures in both $S_1$ and $T_1$ states are depicted in Fig. 3 and Supplementary Figs. 28 and 29 for the type I orientation. The corresponding HOMO and LUMO in $S_1$ (or $T_1$) state are calculated to have 0.12 and 0.13% (0.13 and 0.13% in $T_1$) contribution from PO-T2T and CN-Cz2, respectively, supporting the slight overlap between HOMO and LUMO for efficient TADF. Conversely, for the type II orientation (see Supplementary Figs. 30 and 31), the electron density in HOMO of PO-T2T and that in LUMO of CN-Cz2 for $S_1$ (or $T_1$) state are fully separated (null overlap). Therefore, the type I exciplex configuration well matches the experimental observation, i.e., the results of TADF and GIXD. We further calculated the associated IR spectra based on the type I exciplex configuration (Supplementary Figs. 32 and 33). Note that the weak interaction between PO-T2T and CN-Cz2 in the exciplex makes partial cancellation of the IR peaks between PO-T2T: CN-Cz2 exciplex and individual PO-T2T. Nevertheless, we could attribute those vibrations associated with PO-T2T and CN-Cz2 motions concurrently to the exciplex modes. The results of several distinct peaks calculated as 1282, 1553, and 1577 cm$^{-1}$ (Supplementary Figs. 34 and 35) match well the transient IR spectra (1319, 1547, and 1570 cm$^{-1}$) obtained experimentally. These vibrations show characteristics in that both molecules exert aromatic C = C stretching in combination with bending motions simultaneously due to the exciplex formation. Note that these IR peaks, together with their relaxation dynamics, have been elaborated above. On the basis of the optimized exciplex $S_1$(CT) structure, we then performed a $S_1$(CT) → ground state vertical transition, followed by full geometry optimization to obtain a local minimum $S_0$(LE)' ground state shown in Fig. 3. According to the long interplanar distance of 4.80 Å, the $S_0$(LE)' state is expected to be weakly bound via van der Waals interaction and tends to relax back to initial CN-Cz2:PO-T2T configuration, which, in a qualitative manner, supports the above proposed kinetics during a TADF cycle.

## Discussion

In summary, we report a record-high cyan exciplex OLED (EQE: 16%) consisting of a cyano-contained carbazole-based electron donor CN-Cz2 and an acceptor PO-T2T codeposited as the solid film. We then snapshot the transient IR spectrum of this highly efficient exciplex. Equally important is the observation of polaron-pair absorption spanning the 1700–3000 cm$^{-1}$ region centered at 2060 cm$^{-1}$. This observation supports the formation of polaron-pairs in the exciplex, which either recombines to give charge-transfer emission or is subject to further dissociation to polarons due to different polarization environment between donor and acceptor. Once fully dissociated, generating free positive and negative polarons (see Supplementary Fig. 14), the recombination for light generation is then prohibited. The result offers us a clue to explain the slightly lower emission yield in the

exciplex-type TADF molecules in common. The polaron-pair absorption band may thus serve as a benchmark for the exciplex type TADF OLEDs to correlate with emission quantum efficiency. The Step-Scan time-resolved FTIR spectroscopy thus offers a universal methodology to gain the structural information and its relaxation dynamics, broadening the horizon for harnessing TADF-type OLEDs.

## Methods
Additional synthesis, crystallographic, spectroscopic data, and other support material are provided in the Supplementary Methods.

**Data availability**. The data that support the findings of this study are available from the corresponding author on reasonable request. The X-ray crystallographic coordinates for the CN-Cz2 reported in this article has been deposited at the Cambridge Crystallographic Data Centre (CCDC) under deposition number CCDC 1547184. These data can be obtained free of charge from The Cambridge Crystallographic Data Centre via www.ccdc.cam.ac.uk/data_request/cif.

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

## Acknowledgments

This research was supported by Ministry of Science and Technology (MOST) of Taiwan. We thank Prof. C.-C. Wu and his team for preparing the DMAC-TRZ film. We are also grateful to the National Center for High-performance Computing (NCHC) for the computer time and facilities.

## Author contributions

T.-C.L. developed the transient IR experiment in the solid film. M.S. conducted the synthesis, purification, characterization, and DFT analysis of CN-Cz2. Y.-T.C. did the optical modeling and simulation. S.-H.L. performed the computational calculation. K.-T. L. and P.-Y.C. did the OLED fabrications and photophysical measurements. W.-T.C., Y.-C.L., and H.-F.H. performed GIXD experiments and data analyses. W.-Y.H. designed the OLED structures, analyzed the solid-state and OLED data, and prepared the manuscript. W.-C.T. purified the acceptor PO-T2T. K.-T.W. initiated the molecular design idea for CN-Cz2 and PO-T2T for exciplex formation and prepared the manuscript. P.-T.C. developed the theoretical approach, analyzed, and interpreted the photophysics and prepared the manuscript. All authors discussed the results and contributed to the paper.

## Additional information

**Competing interests:** The authors declare no competing interests.

