## [Peer Review File · Nature Communications]

Reviewers' comments:

Reviewer #1 (Remarks to the Author):

The authors developed an unprecedented approach, pump-probe step-scan Fourier transform IR spectra, to investigate the exciplex system, showing the presence of the polaron-pair states. Although there are still too much noise in the IR decay data, the approach provides some interesting and deep insight on the exciplex states. Considering the different process of optical and electrical excitation, some direct evidence such as that for the polaron-pair states during the electrical excitation could be more helpful for understanding the process of electroluminescence. Anyway, this well-organized manuscript can be recommended for publication, and the developed methodology should be great useful for understanding the structure and relaxation dynamics of TADF OLEDs.

Reviewer #2 (Remarks to the Author):

The novel work of the authors seems to be their method combining the transient FTIR spectroscopy and GIXD to analyze exciplex composed of CN-Cz2 and PO-T2T and to probe their co-deposited film. Through the method they have come to the conclusion that CN-Cz2 and PO-T2T form a lamellar-like packing and a rod-like stacking, respectively. Furthermore, at the interface between the two domains, both exciplex and free polaron pairs can be generated and the generation of polaron pairs is the major cause of the lower PLQY and device efficiency for intermolecular TADF molecules than those for intramolecular TADF molecules.

Their trial and effort to develop the novel combinatorial method to analyze exciplex and intermolecular TADF must be appreciated. However, according to given data in the manuscript and supporting information, it is hard to accept their conclusion. In the Figure 4d, the pi-pi stacking peak of PO-T2T which is shown at about 16 degree in the GIXD profile (green) of the PO-T2T film disappears in the GIXD profile (red) of the CN-Cz2:PO-T2T co-deposited film as described in the manuscript by the authors. According to the authors' explanation, the pi-pi stacking peak disappears due to partial intercalation between CN-Cz2 and PO-T2T. To the best of my knowledge, it tells me that PO-T2T does not form the rod-like stacking any more in the co-deposited film and so there does not exist the interface (illustrated in Figure S13) at which the free polaron pairs can be generated. Furthermore, it is not easy to assign new peaks in red profile and so probably no one can clearly say that there exist interfaces between CN-Cz2 and PO-T2T domains in the red GIXD profile. In addition, it is hard to say that there is the second one of the two broad peaks at 5 and 10 degree because it is too weak (black). Even if we accept the second peak exists, we can hardly find the two broad peaks in the GIXD profile of the co-deposited film any more. It means that we don't have any clear evidence to prove the existence of the interfaces between the CN-Cz2 and PO-T2T domains. I think the authors have to suggest another illustration replacing Figure 4f (and Figure S13) and need to provide revised discussion about the free polaron generation or explain the reason why intermolecular TADF molecules exhibit lower PLQY than intramolecular TADF molecules.

A great number of transient FTIR measurements seem to be performed for this study and through the study with several reference papers, they have concluded the absorption in the region of 1500-2500 cm⁻¹ is the CT state-polaron pair absorption. However to support their explanation of why intermolecular TADF molecules exhibit lower PLQY than intramolecular TADF ones, they need to perform comparative study with intramolecular TADF molecules. With the intramolecular TADF

molecules, there must be no broad peak around 1,700 ~ 3,000 cm⁻¹ or 2,000 ~ 3,000 cm⁻¹. Furthermore, it will be more valuable if it is possible to provide quantitative comparison of free polaron generation between lower-PLQY intermolecular TADF molecules and the CN-Cz2:PO-T2T film with their device results. Without such comparisons, I also hardly understand why the authors reported device results as the former reviewer#1 did.

The novel work the authors developed must be appreciated. However, the gap between their conclusion and the data is not negligible. An alternative model to describe exciplex structure and discussions on it should be provided with comparative studies for publication.

Reviewers' comments:

Reviewer #1 (Remarks to the Author):

The authors developed an unprecedented approach, pump-probe step-scan Fourier transform IR spectra, to investigate the exciplex system, showing the presence of the polaron-pair states. Although there are still too much noise in the IR decay data, the approach provides some interesting and deep insight on the exciplex states. Considering the different process of optical and electrical excitation, some direct evidence such as that for the polaron-pair states during the electrical excitation could be more helpful for understanding the process of electroluminescence. Anyway, this well-organized manuscript can be recommended for publication, and the developed methodology should be great useful for understanding the structure and relaxation dynamics of TADF OLEDs.

Reply: We are very grateful to the reviewer for his/her strong recommendation on this manuscript. We did consider the perhaps different process between optical and electrical excitation. At current stage, the transient FTIR spectra cannot be obtained in the electronic excitation experiments, mainly due to formidable interference from stronger IR absorption of ITO and transporting layers. Nevertheless, the optical process, as supportive by the reviewer as well, should gain many new insights for understanding the structure and relaxation dynamics of TADF OLEDs.

Reviewer #2 (Remarks to the Author):

In the Figure 4f, the pi-pi stacking peak of PO-T2T which is shown at about 16 degree in the GIXD profile (green) of the PO-T2T film disappears in the GIXD profile (red) of the CN-Cz2:PO-T2T co-deposited film as described in the manuscript by the authors. According to the authors' explanation, the pi-pi stacking peak disappears due to partial intercalation between CN-Cz2 and PO-T2T. To the best of my knowledge, it tells me that PO-T2T does not form the rod-like stacking any more in the co-deposited film and so there does not exist the interface (illustrated in Figure S13) at which the free polaron pairs can be generated. Furthermore, it is not easy to assign new peaks in red profile and so probably no one can clearly say that there exist interfaces between CN-Cz2 and PO-T2T domains in the red GIXD profile.

Reply:

We appreciate reviewer's valuable comment. All three 1D GIXD plots were fitted carefully, although the fitting results were not included in original SI. All the deconvoluted signals of the co-deposited films were traced by comparing with line shapes (d-spacing and line width) of signals of individual components.

We apologize that we did not clearly describe the GIXD changes of the films in the first submission. In the co-deposited film, the signal at ca. 16 degree (3.8

angstrom) of individual **PO-T2T** does disappear. However, the π - π stacking peak of **PO-T2T** in the co-deposited film is shifted to a smaller angle, i.e., a larger d-spacing (4.2 angstrom) with respect to the corresponding signal of the individual component. Moreover, from the deconvolution data, the rod-like stacking of **PO-T2T** is a bit shorter in the co-deposited film than in the pure **PO-T2T**. In short, the rod-like stacking of **PO-T2T** retains, in a qualitative manner, but with a slightly larger intermolecular separation. On the other hand, the cluster of **CN-Cz2** in the co-deposited film is also affected, which is evidenced by the out-of-plane correlation signal, shifting from 4.27 angstrom to 4.32 angstrom, i.e., a larger spacing. The inter-planar spacing changes of both components in the co-deposited film are consistent with the mutual intercalation structure. We have revised the discussion of the GIXD data accordingly (see page 11, lines 9-13 of the revised manuscript).

In addition, it is hard to say that there is the second one of the two broad peaks at 5 and 10 degree because it is too weak (black). Even if we accept the second peak exists, we can hardly find the two broad peaks in the GIXD profile of the co-deposited film any more. It means that we don't have any clear evidence to prove the existence of the interfaces between the CN-Cz2 and PO-T2T domains.

Reply:

We are grateful to the reviewer for his/her valuable comments. Again, all three 1D GIXD plots were fitted carefully. We are sorry that the fitting results were not included in original SI. All the deconvoluted signals of co-deposited film were traced by comparing with line shapes (d-spacing and line width) of signals of each individual component (see Figures S24 and S25 of the supporting information).

For **CN-Cz2**, the fitted 1D GIXD plot (shown below) confirms the second broad peak (ca. 10 degree).

Figure S24. The fitted 1D GIXD of **CN-Cz2** film.

For the co-deposited film, the fitted 1D GIXD plot is also shown below. The first broad signal at 12.2 angstrom (ca. 5 degree) of **CN-Cz2** is shifted to 12.6 angstrom in the co-deposited film. The second broad signal (ca. 10 degree) is likely buried under the intense peak at ca. 9.5 degree.

Figure S25. Fitted 1D GIXD of co-deposited film

I think the authors have to suggest another illustration replacing Figure 4f (and Figure S13) and need to provide revised discussion about the free polaron generation or explain the reason why intermolecular TADF molecules exhibit lower PLQY than intramolecular TADF molecules.

Reply:

We are grateful to the reviewer for his/her valuable suggestions. According to the above revisions on GIXD, we have redrawn Figure 4f and merged 4e/4f to clearly show the mutual intercalation of the two components evidenced by GIXD studies (see below and the revised Figure 4).

Figure 4. 2D GIXD patterns for **a.** CN-Cz2, **c.** CN-Cz2:PO-T2T (1:1) and **e.** PO-T2T and the corresponding 1D GIXD patterns **b.**, **d.**, and **f.**, respectively. **g.** The molecular packing for CN-Cz2 grain domain, PO-T2T grain domain, and the in-plane interaction between CN-Cz2 and PO-T2T grain domains. Note that the 1D GIXD profile of PO-T2T shows a strong diffraction peak at $2\theta = 15.5^\circ$, corresponding to a well correlated π - π stacking with a d-spacing of 3.8 Å as illustrated in **g.** In CN-Cz2:PO-T2T (1:1) co-deposited film, the diffraction corresponding to the π - π stacking of PO-T2T shifts to a larger d-spacing as indicated by the deconvolution analyses. The interplanar spacings of the two components in the co-deposited film are both larger than those in films of individual components, indicating interaction of the two components in the co-deposited film as depicted in **g** (right).

A great number of transient FTIR measurements seem to be performed for this study and through the study with several reference papers, they have concluded the absorption in the region of 1500-2500 cm⁻¹ is the CT state-polaron pair absorption. However to support their explanation of why intermolecular TADF molecules exhibit lower PLQY than intramolecular TADF ones, they need to perform comparative study with intramolecular TADF molecules. With the intramolecular TADF molecules, there must be no broad peak around 1,700 ~ 3,000 cm⁻¹ or 2,000 ~ 3,000 cm⁻¹.

Reply:

We are grateful to the reviewer for his/her valuable comments and suggestions. Accordingly, we have carried out the comparative step-scan FTIR experiments for three intramolecular TADF molecules (PXZ-TRZ, DMAC-TRZ and PXZPBM, see Figure S16-S20 of the supporting information) under the same excitation power and similar optical absorbance for the prepared film. As a result, we successfully acquired transient FTIR for these three intramolecular TADF molecules (see Figure S16-S20). First of all, the success proves the applicability of step-scan FTIR to all intra- and inter-molecular types of TADF cases. More importantly, our results indicate that there are none of the intramolecular TADF molecules for which the polaron-pair absorption could be observed in the range of 1,700-3,000 cm⁻¹. This is reasonable because the strong bound excitons lead to negligible polaron-pair formation in the intramolecular TADF cases.

Furthermore, it will be more valuable if it is possible to provide quantitative comparison of free polaron generation between lower-PLQY intermolecular TADF molecules and the CN-Cz2:PO-T2T film with their device results. Without such comparisons, I also hardly understand why the authors reported device results as the former reviewer#1 did.

The novel work the authors developed must be appreciated. However, the gap between their conclusion and the data is not negligible. An alternative model to describe exciplex structure and discussions on it should be provided with comparative studies for publication.

According to the results of CN-Cz2:PO-T2T film, the decrease of exciplex type TADF emission, in part, correlates with the polaron generations. For generalization, we also performed other types of exciplex, for example NPNPB (donor) and PO-T2T (see Figure S15), which gave inferior exciplex emission yield and OLED performance (Hung, W.-Y., et al. "The First Tandem, All-exciplex-based WOLED" Sci. Rep. 4, 5161 (2014)). The time-resolved FTIR results shown below and Figure S15 in SI obviously reveal a broad transient polaron-pair absorption band extended to > 3000 cm⁻¹. Its higher

frequency than that of CN-Cz2:PO-T2T may indicate the more stable and hence efficient polaron-pair formation, which is subsequently subject to the polaron dissociation. This trend may offer us a clue to explain the lower emission yield in the exciplex-type TADF molecules in common, that is, in a semi-quantitative manner, the decrease of TADF emission, in part, correlates with the polaron-pair generations. This proposed mechanism is novel and should impact on the TADF relevant research fields.

Figure S15. a. Chemical structures of NPNPB and PO-T2T. b. An absorbance-frequency-time 3D plot for the NPNPB:PO-T2T (1:1). Excitation wavelength: 266 nm. c. The transient IR spectra of NPNPB:PO-T2T film at different delay times in spectral range of 1000-3000 cm^{-1} . d. The relaxation dynamics of 1257 and 1550 cm^{-1} , e. 1512 and 1527 cm^{-1} and f. 2700 and 2800 cm^{-1} polaron-pair absorption band for the NPNPB:PO-T2T film. The PL decay lifetime is about 180 ns.

REVIEWERS' COMMENTS:

Reviewer #1 (Remarks to the Author):

The authors revised their manuscript carefully and addressed almost all the issues. It can now be recommended for publication without further revision.